# Recurrent disruption of tumour suppressor genes in cancer by somatic mutations in cleavage and polyadenylation signals

Yaroslav Kainov[1,2]*, Fursham Hamid[1], Eugene V Makeyev[1]*

[1]Centre for Developmental Neurobiology, King's College London, London, United Kingdom; [2]Department of Medical and Molecular Genetics, King's College London, London, United Kingdom

## eLife Assessment

This **important** study substantially advances our understanding of noncoding somatic mutations by identifying a novel class of mutations that affect 3'UTR polyadenylation signals enriched in tumor suppressor genes in cancer. The evidence supporting the conclusions is **convincing**, with rigorous statistical analyses. The work will be of broad interest to cancer researchers.

*For correspondence:
yaroslav.kainov@kcl.ac.uk (YK);
eugene.makeyev@kcl.ac.uk
(EVM)

**Competing interest:** The authors declare that no competing interests exist.

**Abstract** The expression of eukaryotic genes relies on the precise 3'-terminal cleavage and polyadenylation of newly synthesized pre-mRNA transcripts. Defects in these processes have been associated with various diseases, including cancer. While cancer-focused sequencing studies have identified numerous driver mutations in protein-coding sequences, noncoding drivers – particularly those affecting the cis-elements required for pre-mRNA cleavage and polyadenylation – have received less attention. Here, we systematically analysed somatic mutations affecting 3'UTR polyadenylation signals in human cancers using the Pan-Cancer Analysis of Whole Genomes (PCAWG) dataset. We found a striking enrichment of cancer-specific somatic mutations that disrupt strong and evolutionarily conserved cleavage and polyadenylation signals within tumour suppressor genes. Further bioinformatics and experimental analyses conducted as a part of our study suggest that these mutations have a profound capacity to downregulate the expression of tumour suppressor genes. Thus, this work uncovers a novel class of noncoding somatic mutations with significant potential to drive cancer progression.

## Introduction

Most eukaryotic mRNAs are modified by the addition of a 5'-terminal 7-methylguanosine cap, splicing of intronic sequences, and 3'-terminal cleavage and polyadenylation. The cleavage and polyadenylation reactions are tightly coupled with transcription termination and the release of newly synthesized transcripts from RNA polymerase II. Therefore, precise cleavage and polyadenylation are critical for the production of mature mRNAs. Mechanistically, these reactions require a co-transcriptional assembly of a multisubunit protein complex at the corresponding cis-regulatory sequences near the pre-mRNA cleavage site (CS) (*Tian and Manley, 2017*; *Neve et al., 2017*; *Shi et al., 2009*). A key cis-element guiding the assembly of the cleavage and polyadenylation machinery is the polyadenylation signal (PAS). The most common PAS sequences in mammals are AATAAA and ATTAAA hexamers,

although their single nucleotide-substituted variants may function in some cases (*Proudfoot, 1991*; *Beaudoing et al., 2000*).

Earlier studies have emphasized the importance of pre-mRNA cleavage/polyadenylation in the context of human diseases. For example, alternative cleavage/polyadenylation has been proposed to modulate the expression of oncogenes and tumour suppressors in different types of cancer (*Mayr and Bartel, 2009*; *Lee et al., 2018*). Germline mutations affecting polyadenylation signals can play a role in genetic disorders (*Higgs et al., 1983*; *Bogard et al., 2019*; *Bennett et al., 2001*) and increase cancer susceptibility (*Stacey et al., 2011*; *Li et al., 2023*). Notably, although the role of somatic mutations affecting polyadenylation signals has been investigated for individual genes in a limited number of tumour samples (*Wiestner et al., 2007*; *Shlien et al., 2016*), a systematic characterisation of the role of this type of mutation in cancer has not been carried out.

Large-scale genome sequencing studies have identified numerous cancer driver and driver-like mutations within protein-coding sequences (*ICGC/TCGA Pan-Cancer Analysis of Whole Genomes Consortium, 2020*). Such mutations have also been mapped to noncoding regions; however, existing research has primarily focused on promoters, enhancers, and splicing signals (*Cao et al., 2020*; *Zhao et al., 2021*; *Sherman et al., 2022*; *Rheinbay et al., 2020*; *Calabrese et al., 2020*), rather than sequences regulating pre-mRNA cleavage and polyadenylation.

Here, we conducted a systematic genome-wide analysis of somatic single-nucleotide variants (SNVs) affecting the PAS elements in mRNA 3'untranslated regions (3'UTRs) in cancer cells. Using a large tumour whole-genome sequencing dataset, the PCAWG (*ICGC/TCGA Pan-Cancer Analysis of Whole Genomes Consortium, 2020*), we found that strong and evolutionarily conserved cleavage/polyadenylation signals are often disrupted by cancer-specific SNVs. Strikingly, such mutations are significantly enriched in tumour suppressor genes. We further provide evidence that such mutations can substantially decrease the expression of tumour suppressor genes in cancer cells. Overall, our work identifies a novel class of noncoding somatic mutations with driver-like properties in cancer.

## Results
### Somatic mutations often disrupt cleavage and polyadenylation sequences in cancer

We first analysed SNVs neighbouring annotated human cleavage and polyadenylation positions (paSNVs) in 3'UTRs from the PolyA_DB3 database (*Wang et al., 2018*). We considered two distinct cohorts: 'Normal' paSNVs from a healthy human population (the 1000 Genomes phase 3 data *Fairley et al., 2020*) and 'Cancer' paSNVs from the whole-genome sequencing of cancer samples (PCAWG) (*ICGC/TCGA Pan-Cancer Analysis of Whole Genomes Consortium, 2020*).

For each paSNV, we calculated the change in cleavage/polyadenylation efficiency using the APARENT2 neural network model, which has been shown to infer this statistic more accurately than earlier approaches (*Linder et al., 2022*). We additionally assessed the loss and gain of the two strongest polyadenylation signals, AATAAA and ATTAAA (referred to as AWTAAA throughout this study; *Figure 1A*).

As expected, paSNVs predicted to have a strong impact on cleavage/polyadenylation were often situated immediately upstream of a CS, with most of them affecting AWTAAA hexamers (*Figure 1B*). Furthermore, the presence of AATAAA or ATTAAA in polyadenylation signals tended to be associated with a high APARENT2 score (*Figure 1—figure supplement 1A*), and the loss or gain of AATAAA led to a significantly stronger decrease or increase in the score, respectively, compared to other mutations (*Figure 1—figure supplement 1B*). This provided internal validation of the algorithm's performance in our hands.

We then categorized all paSNVs into three groups: (1) upregulating cleavage/polyadenylation (UP-paSNVs), defined as events with a ≥1 increase in the APARENT2 score (log odds ratio, or LOR) and creating an AWTAAA hexamer; (2) downregulating cleavage/polyadenylation (DOWN-paSNVs; LOR≤–1 and disrupting an AWTAAA); and (3) the remaining annotated cleavage position-adjacent paSNVs (*Figure 1B*). The latter group served as a background control (BG-paSNVs) in our subsequent analyses.

Consistent with the earlier studies (*Linder et al., 2022*; *Kainov et al., 2016*; *Findlay et al., 2022*), we observed a pronounced negative selection against the DOWN-paSNVs in the Normal dataset

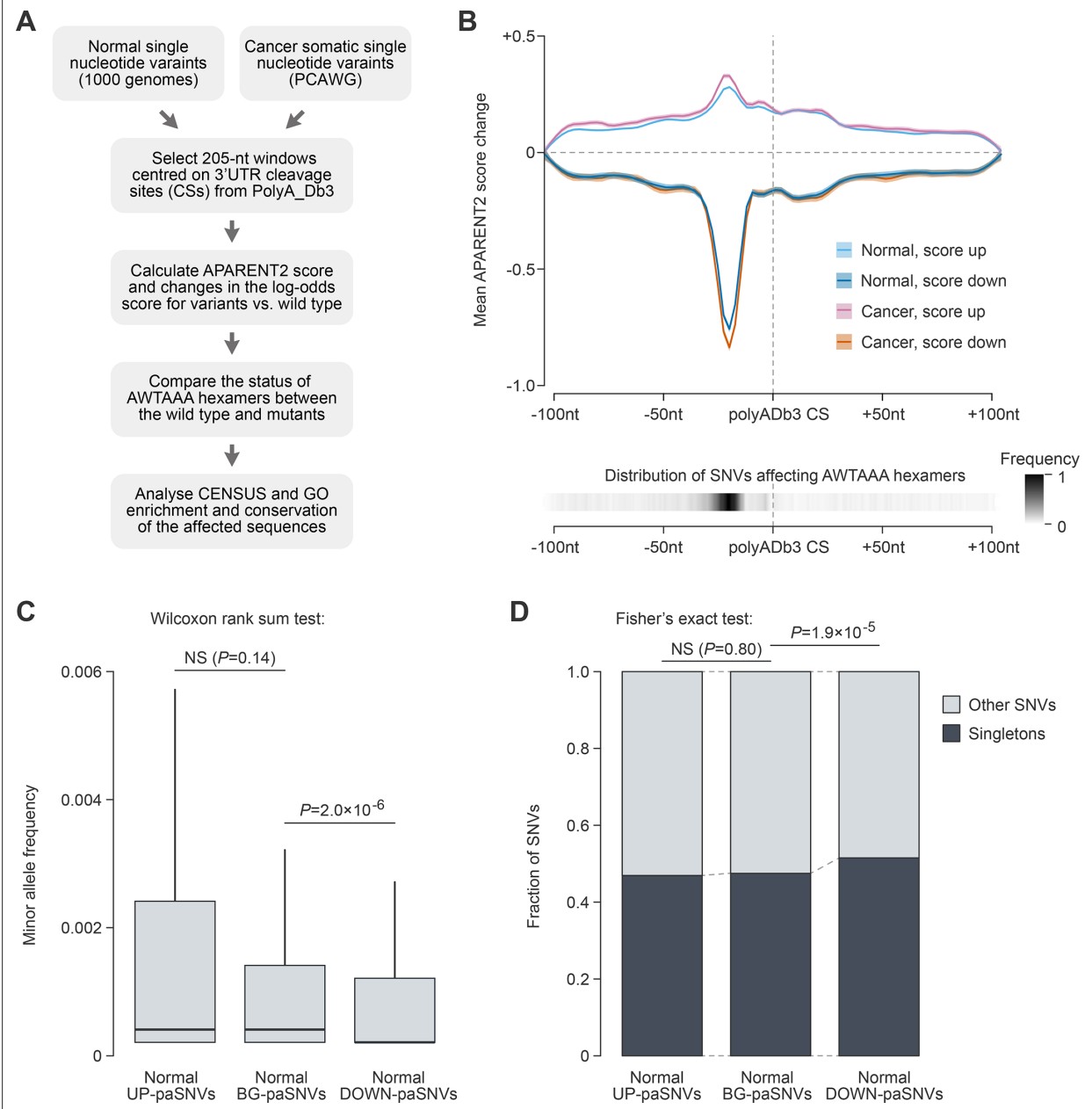

**Figure 1.** paSNVs disrupting cleavage/polyadenylation signals are depleted in the normal population. (**A**) Bioinformatics workflow used to analyse the effect of paSNVs on pre-mRNA cleavage and polyadenylation. (**B**) Top, effects of UP- and DOWN-paSNVs on the APARENT2 score (mean ± SEM) as a function of their position with respect to annotated pre-mRNA cleavage sites (CSs). Bottom, combined distribution of AWTAAA-affecting paSNVs in both datasets. (**C**) Box plot showing that paSNVs disrupting polyadenylation signals are significantly less frequent compared to control groups of events in the normal population. (**D**) paSNVs disrupting polyadenylation signals are enriched for singletons, consistent with purifying selection against such events in the normal population.

The online version of this article includes the following figure supplement(s) for figure 1:

**Figure supplement 1.** Distribution of cleavage/polyadenylation signal-disrupting mutations in the normal population (1000 genomes dataset).

(*Figure 1C, D*). This category showed significantly decreased allele frequencies in comparison to the BG-paSNVs and was enriched for singletons (unique variants in the analysed dataset). This effect was more evident when considering changes in both the score and the hexamer composition (*Figure 1— figure supplement 1C, D*).

Notably, a comparison of the Normal and Cancer datasets showed that cancer somatic mutations, on average, had a stronger effect on the polyadenylation efficiency in both the UP- and DOWN-paSNV

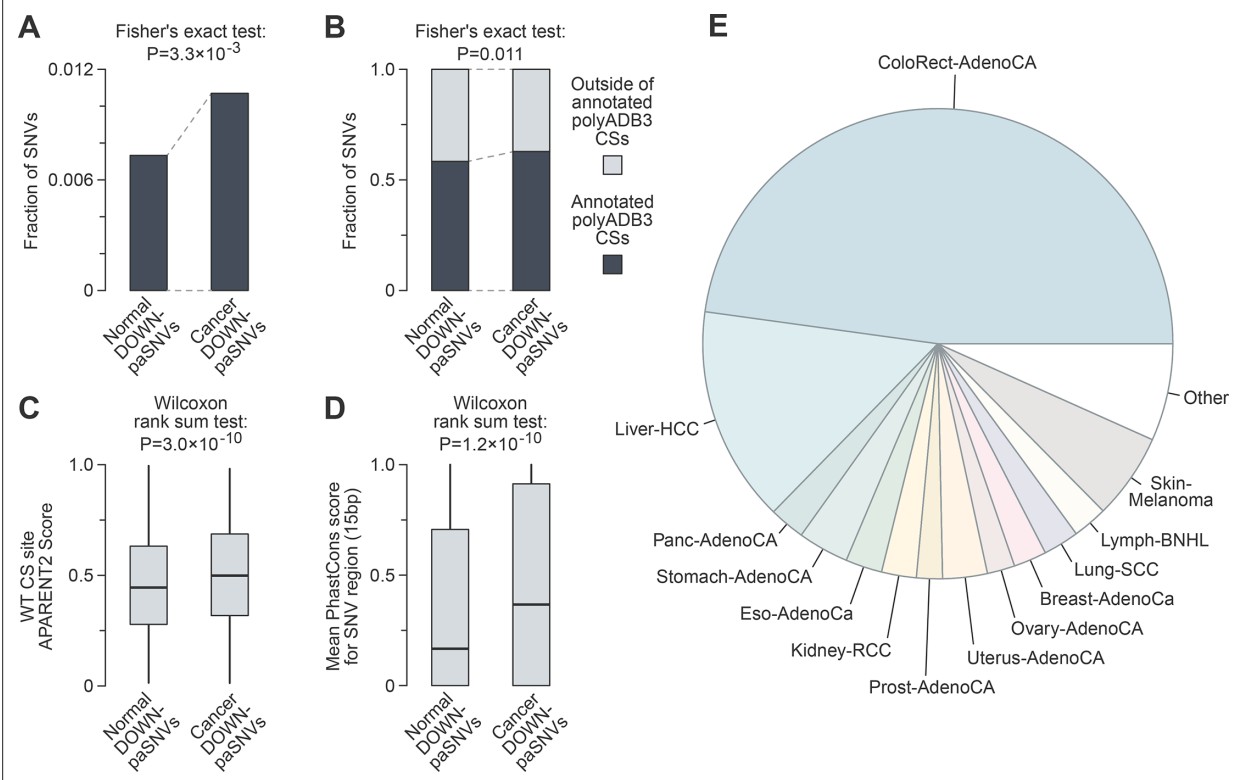

**Figure 2.** Cancer somatic mutations tend to disrupt functional cleavage/polyadenylation signals. (**A**) Bar plot showing enrichment of paSNVs disrupting polyadenylation signals among cancer somatic mutations. (**B**) Bar plot showing enrichment of single-nucleotide variants (SNVs) affecting AWTAAA sequences in 3'UTRs close to annotated cleavage sites (CSs) among cancer somatic mutations. (**C**) Box plot showing that somatic mutations disrupt stronger cleavage/polyadenylation signals in cancer. (**D**) paSNVs disrupting polyadenylation signals occur in more evolutionary conserved regions in cancer (mean PhastCons score in 15-nt window centred at SNVs). (**E**) Distribution of DOWN-paSNVs across cancer types in the Pan-Cancer Analysis of Whole Genomes (PCAWG) project.

The online version of this article includes the following figure supplement(s) for figure 2:

**Figure supplement 1.** Cancer somatic DOWN-paSNVs often occur in evolutionarily conserved regions.

groups (*Figure 1B*; LOR sample variance 0.115 in cancer vs 0.0876 in normal). DOWN-paSNVs were significantly enriched in cancer compared to the normal population data (*Figure 2A*). We also observed that mutations disrupting AWTAAA hexamers in 3'UTRs tended to occur near annotated cleavage sites in cancer (*Figure 2B*).

Interestingly, cancer-specific DOWN-paSNVs affected cleavage/polyadenylation signals with higher APARENT2 scores (*Figure 2C*). Furthermore, DOWN-paSNVs tended to affect more evolutionarily conserved sequences in the Cancer dataset compared to the Normal control (*Figure 2D*, *Figure 2—figure supplement 1*). In total, we identified 1614 distinct cancer somatic DOWN-paSNVs affecting 1570 cleavage/polyadenylation events in 1460 genes in 602 tumours, i.e., 22.7% of all tumour samples in PCAWG (*Supplementary file 1*). Notably, nearly half of the DOWN-paSNVs originated from colorectal adenocarcinoma, with the remaining mutations distributed across a wide range of other cancer types (*Figure 2E*).

We concluded that mutations disrupting functional cleavage/polyadenylation signals are abundant in cancer cells despite being subject to strong purifying selection in a healthy population.

## Cancer-specific mutations in cleavage and polyadenylation sequences are enriched in tumour suppressor genes

There are two possible explanations for the enrichment of DOWN-paSNV events in cancer: (1) an increase in the overall mutation load and (2) positive selection for such mutations. Since the latter possibility may increase the incidence of mutations in cancer driver genes, we analysed the distribution

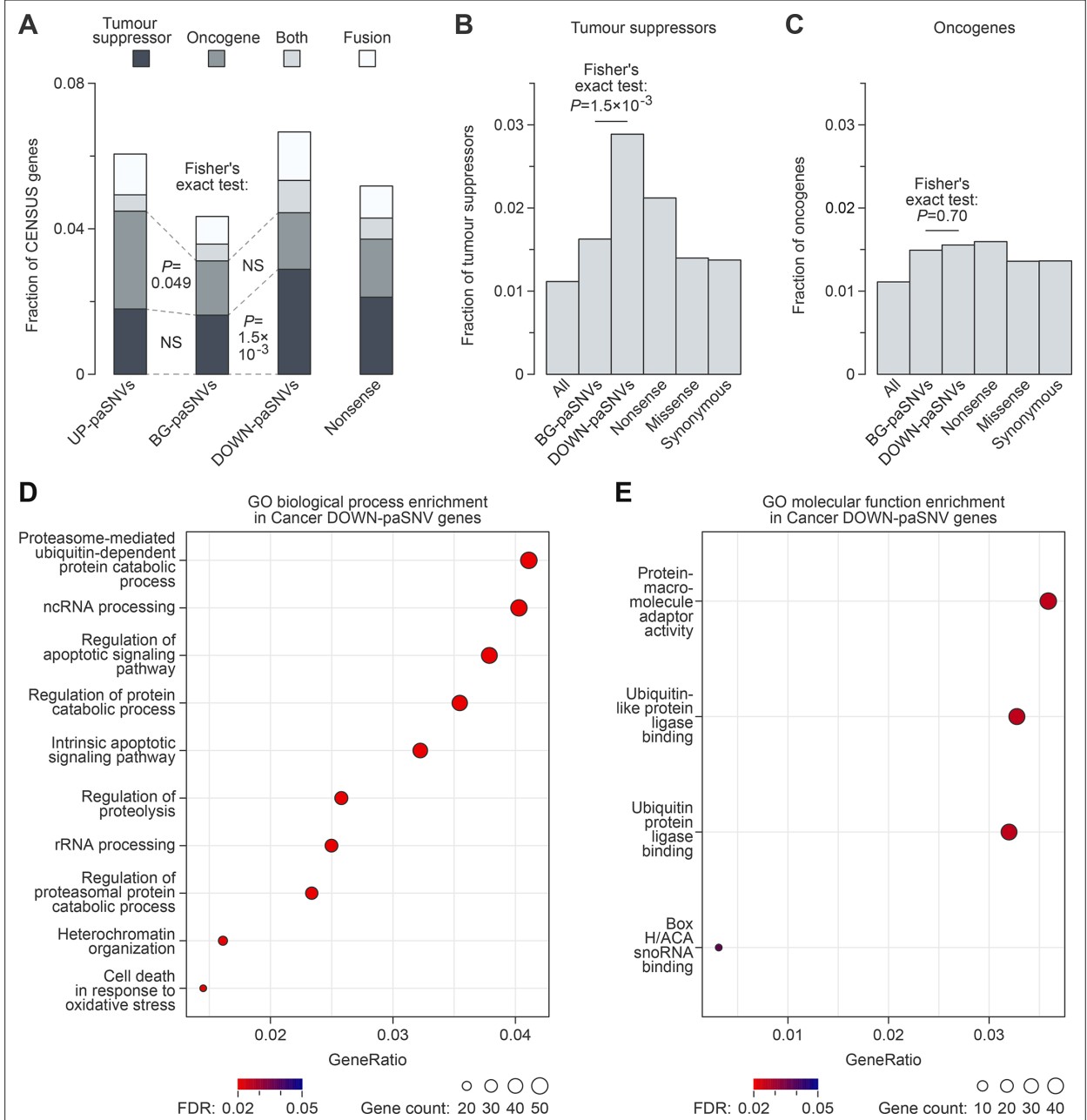

**Figure 3.** Somatic cancer mutations often disrupt cleavage/polyadenylation signals in tumour suppressor genes. (**A**) Stacked bar plot showing enrichment of single-nucleotide variants (SNVs) disrupting polyadenylation signals (DOWN-paSNVs) in tumour suppressors in cancer. (**B–C**) Overrepresentation of (**B**) tumour suppressors but not (**C**) oncogenes among genes with cancer somatic DOWN-paSNVs, as compared to genes with cancer somatic BG-paSNVs. Fractions of tumour suppressors and oncogenes are also shown for all genes and genes containing cancer somatic nonsense (premature stop codons), missense (altered amino acid residues) and synonymous (synonymous codons) mutations. Note that the enrichment of tumour suppressors is stronger for DOWN-paSNVs compared to nonsense mutations. (**C**) Top 10 GO Biological Process terms significantly enriched in genes with cancer somatic DOWN-paSNVs. Note the enrichment of apoptosis- and cell death-related functions. (**D**) GO Molecular Function terms significantly enriched in genes with cancer somatic DOWN-paSNVs.

The online version of this article includes the following figure supplement(s) for figure 3:

**Figure supplement 1.** Cancer somatic DOWN-paSNVs often reside in genes with tumour-suppressive functions.

of DOWN-paSNVs within genes from the Cancer Gene Census (*Tate et al., 2019*). This revealed a remarkable over-representation of the DOWN-paSNVs in tumour suppressor genes, with the magnitude of this effect being greater than the corresponding enrichment of nonsense mutations (SNVs creating a premature translation termination codon) (*Figure 3A, B*, *Figure 3—figure supplement 1*).

Notably, DOWN-paSNVs were not enriched in the oncogenes (*Figure 3C*), in line with the disruptive nature of such mutations under normal conditions (*Figure 1*). Conversely, oncogenes but not tumour suppressors showed some enrichment for UP-paSNVs (*Figure 3A* and *Figure 3—figure supplement 1*).

Overall, DOWN-paSNVs were found to affect 38 tumour suppressor genes, i.e., 14.3% of all genes in this category in the Census dataset. In several cases, including *LRP1B* and *FOXO1*, which are known to act as tumour suppressors in certain cancers, the same signal/polyadenylation signal was disrupted by the same or different mutations in more than one sample (see columns Mut_Recurrence and Signal_Recurrence in *Supplementary file 1*). Consistent with tumour suppressors being a major target of DOWN-paSNVs, genes with this type of mutations were significantly enriched for apoptosis-related functions (*Figure 3D*; e.g. tumour suppressors *CASP9* or *FHIT*). We also detected the enrichment of proteins interacting with the ubiquitination pathway (*Figure 3D, E*; e.g. tumour suppressors *SMAD2*, *APC,* and *AXIN1*).

To independently confirm the functional impact of DOWN-paSNVs in cancer, we compared the mutational excess of different types of somatic mutations using DigDriver (*Sherman et al., 2022*), a neural network-based method that accounts for cancer-specific mutation rates. This analysis revealed a significantly higher observed-to-expected mutation rate for DOWN-paSNV events in cancer compared to the BG-paSNV group (*Figure 4—figure supplement 1*). DOWN-paSNVs tended to be enriched in tumour suppressor genes, consistent with positive selection for these events in cancer (*Figure 4A*). No such enrichment was detected in oncogenes (*Figure 4B*).

Of note, our analysis of wild-type sequences showed that tumour suppressor 3'UTRs are characterized by stronger cleavage/polyadenylation signals compared to oncogenes and non-cancer genes (*Figure 4—figure supplement 2A, B*). Moreover, tumour suppressors associated with hallmarks of cancer (*Hanahan and Weinberg, 2011*) in the Census dataset had stronger cleavage/polyadenylation signals than the rest of tumour suppressor genes (*Figure 4—figure supplement 2C*).

According to the classical two-hit hypothesis (*Knudson, 1971*), both alleles of tumour suppressor genes may acquire distinct damaging mutations in cancer. With this in mind, we analysed the co-occurrence of paSNVs with damaging non-synonymous mutations from the PCAWG collection (non-syn. variants from the binarized gene-centric table in *Calabrese et al., 2020*). DOWN-paSNV-containing tumour suppressors showed a markedly increased incidence of such additional somatic mutations in the same tumour compared to the BG-paSNV control (*Figure 4C*). Furthermore, the overall frequency of damaging non-synonymous mutations in tumour suppressors affected by DOWN-paSNVs in at least one sample was significantly higher than in the DOWN-paSNV-negative tumour suppressor group (*Figure 4D*).

The analysis of allele copy number variation (CNV) showed that increased copy number was 4.1 times more common in the PCAWG data compared to allele loss. However, the incidence of copy number increase was substantially lower in the DOWN-paSNV group compared to the BG-paSNV control (*Figure 4—figure supplement 3*). This points to a negative selection against duplications of genes affected by DOWN-paSNVs in cancer.

Taken together, these data suggest that somatic mutations disrupting cleavage and polyadenylation can facilitate the inactivation of tumour suppressors in cancer.

## Somatic mutations in cleavage and polyadenylation signals can decrease the expression of tumour suppressor genes

Genetic inactivation of functional cleavage/polyadenylation sequences may negatively affect gene expression (see e.g. *Higgs et al., 1983*). To explore this possibility, we turned to the colorectal adenocarcinoma subset of PCAWG, as it contained most of the DOWN-paSNVs in tumour suppressors and the corresponding gene expression information (*Calabrese et al., 2020*). We shortlisted detectably expressed tumour suppressors that contained DOWN-paSNVs and no other damaging mutations in specific cancer samples, and were wild-type in other samples. Seven genes passing these filters were involved in various aspects of tumour biology, including cell survival and DNA

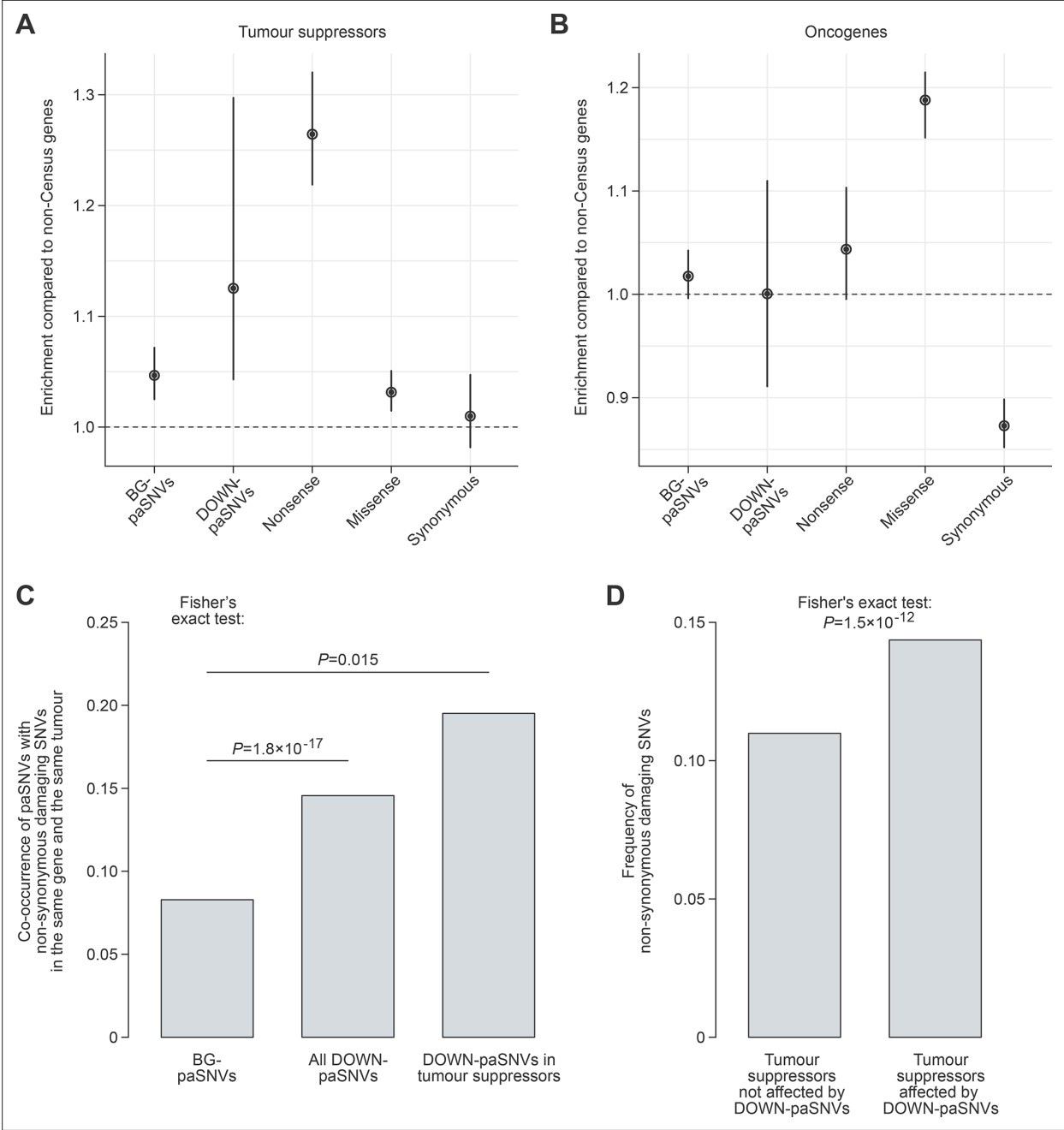

**Figure 4.** Disruption of cleavage/polyadenylation signals in tumour suppressors, along with other damaging mutations, may facilitate cancer progression. (**A–B**) Enrichment of different groups of cancer somatic single-nucleotide variants (SNVs) in (**A**) tumour suppressors and (**B**) oncogenes calculated using DigDriver relative to genes not listed in Cancer Census (non-Census) and presented with 95% confidence intervals. Note that DOWN-paSNVs and nonsense mutations are enriched in tumour suppressors but not in oncogenes. In contrast, oncogenes are often affected by missense mutations, as expected. (**C**) Cancer somatic DOWN-paSNVs co-occur in the same tumour with non-synonymous damaging SNVs, a group of somatic mutations defined in *Calabrese et al., 2020*, more often than BG-paSNVs. Note that the co-occurrence is particularly high for tumour suppressors. (**D**) The overall frequency of non-synonymous damaging SNVs is significantly higher in the DOWN-paSNV-containing group compared to the DOWN-paSNV-lacking group of tumour suppressor genes.

The online version of this article includes the following figure supplement(s) for figure 4:

**Figure supplement 1.** Cancer somatic DOWN-paSNVs are enriched for statistically significant DigDriver events (BH-adjusted p<0.01), suggesting that they may be under positive selection in cancer.

**Figure supplement 2.** Wild-type tumour suppressor genes tend to have efficient cleavage/polyadenylation signals.

**Figure supplement 3.** Reduced tendency for copy number increases in genes with DOWN-paSNVs.

repair (*CASP9*, *NDRG1*, and *XPA*), mTOR signalling (*TSC1*), and transcription and RNA processing (*ETV6*, *ISY1,* and *SMAD2*).

Plotting pairwise gene-specific expression differences for the aggregated tumour suppressor set, we observed a significant bias towards downregulation in the samples containing DOWN-paSNVs compared to the wild-type controls (*Figure 5A*; median downregulation of 1.25-fold). Remarkably, similar negative biases were detected for all seven individual genes, with median downregulation values ranging from 1.1- to 3.2-fold (*Figure 5B*).

To validate the effect of DOWN-paSNVs on gene expression, we focused on a somatic mutation that disrupts the cleavage/polyadenylation signal in the tumour suppressor *XPA*. This gene has been shown to promote apoptosis in response to DNA damage, in addition to its role in nucleotide excision repair (*Deng et al., 2021*). Moreover, downregulation of *XPA* has been associated with decreased patient survival in colorectal cancer (*Feng et al., 2018*).

The *XPA* mutation identified by our bioinformatics analyses alters the canonical AATAAA PAS hexamer to GATAAA near the terminal CS and significantly reduces the APARENT2 score (*Figure 5C*). To experimentally assess the effect of this mutation on the efficiency of pre-mRNA cleavage and polyadenylation, we prepared minigene constructs where the wild-type or mutant sequences were inserted upstream of a recombinant CS (*Figure 5D*).

We used the wild-type and mutant minigenes to transfect the human colorectal cancer cell line HCT-116. An RT-qPCR assay measuring the efficiency of cleavage/polyadenylation as a ratio between the CS-read-through and CS-upstream signals revealed a significant decrease in cleavage/polyade-nylation efficiency in response to the *XPA* DOWN-paSNV (*Figure 5D*).

To directly assess the effect of defective cleavage/polyadenylation on gene expression, the wild-type or the mutant 3'UTR sequences were inserted downstream of a luciferase reporter gene. Following the transfection of HCT-116 cells, we detected a significantly reduced production of luciferase protein from the mutant construct compared to the wild-type control (*Figure 5E*).

Thus, somatic mutations disrupting polyadenylation signals in tumour suppressor genes can reduce the abundance of functional mRNA transcripts.

## Discussion

We interrogated whole-genome mutation data using recently developed machine-learning approaches to systematically characterize the impact of SNVs on 3'UTR polyadenylation signals (PAS) in cancer. Our analyses confirm that germline SNVs disrupting PAS are likely deleterious, as they are subjected to strong negative selection in the normal population (*Figure 1*). Intriguingly, we found that somatic mutations affecting such cis-elements in cancer are more prevalent, tend to occur near stronger CSs, and target more evolutionarily conserved PAS hexamers (*Figure 2*, *Figure 2—figure supplement 1*).

Importantly, these cancer somatic SNVs disrupt PAS sequences in tumour suppressor genes with a similar enrichment pattern to well-known deleterious SNVs in protein-coding regions, such as nonsense mutations (*Figures 3 and 4*). Additionally, wild-type tumour suppressors have stronger cleavage/polyadenylation signals than other groups of genes (*Figure 4—figure supplement 2*), pointing to the importance of the corresponding steps of pre-mRNA processing for their expression.

Consistent with the two-hit hypothesis (*Knudson, 1971*), we found that tumour suppressors with disrupted cleavage/polyadenylation signals (i.e. containing DOWN-paSNVs) are more likely to acquire other damaging somatic mutations in the same tumour (*Figure 4C, D*). As long-read genomic sequencing data become increasingly available, it will be interesting to investigate whether these additional mutations occur in the same or in a different allele compared to the DOWN-paSNVs.

However, it is possible that DOWN-paSNVs can contribute to tumour progression even in the absence of other mutations. Indeed, tumour suppressors containing only DOWN-paSNVs are consistently expressed at lower levels compared to their wild-type counterparts (*Figure 5A, B*). Moreover, it is currently thought that partial inactivation of many tumour suppressors can be sufficient to promote tumorigenesis (*Berger et al., 2011*; *Park et al., 2021*).

Using the tumour suppressor gene *XPA* as an example, we directly show that a cancer-specific single-nucleotide mutation disrupting the PAS hexamer is sufficient to block pre-mRNA cleavage/polyadenylation and dampen the expression of mature mRNA (*Figure 5C, E*). These results support our bioinformatics analyses and argue that SNVs targeting polyadenylation signals can have a profound

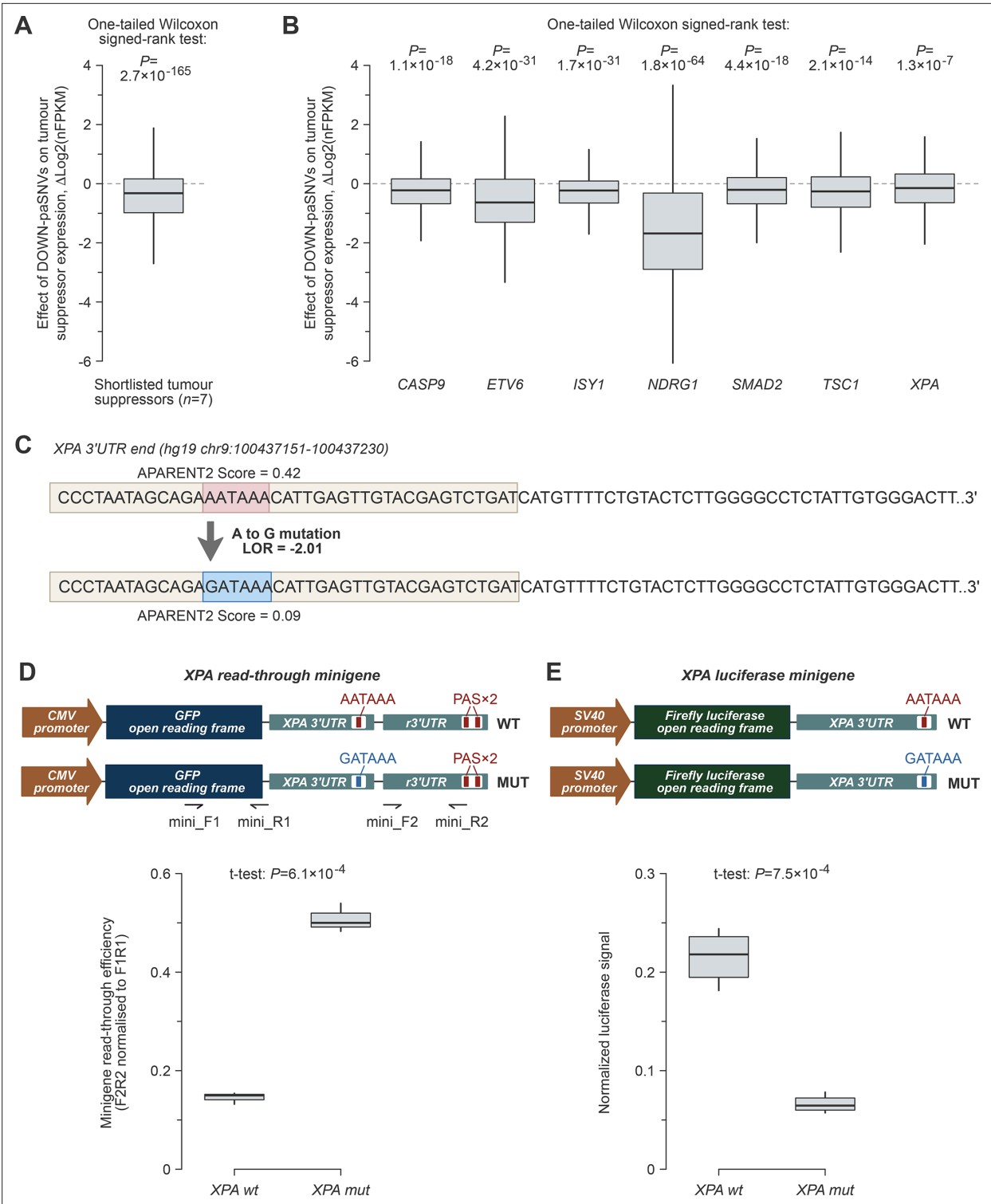

**Figure 5.** Somatic cancer DOWN-paSNVs are sufficient to downregulate tumour suppressor genes. (**A, B**) Gene-specific expression differences between DOWN-paSNV-containing and wild-type samples (ΔLog2 of copy number variation-normalized FPKM values; see Materials and Methods) reveal a consistently negative effect of DOWN-paSNV on tumour suppressor mRNA abundance in colorectal cancers. Box plots are shown for (**A**) an aggregated set of qualifying tumour suppressors and (**B**) individual genes from this set. Outliers are omitted for clarity. (**C**) Wild-type and mutated sequences of the *XPA* tumour suppressor gene cleavage/polyadenylation signal. The polyadenylation signal (PAS) hexamer is enclosed within a box. (**D**) Top, *XPA* cleavage site read-through minigenes and corresponding primers used for RT-qPCR analyses. Bottom, RT-qPCR data showing stronger read-through (weaker polyadenylation) in the mutant minigene. (**E**) Top, luciferase expression minigenes. Bottom, luciferase assay reveals that the cancer-specific PAS mutation dampens the expression of the reporter gene.

effect on gene expression in cancer. Our data are also consistent with previous reports showing similar gene expression effects of PAS-specific germline SNVs (*Higgs et al., 1983*; *Bennett et al., 2001*).

It is expected that mutation of cleavage/polyadenylation signals should lead to the appearance of abnormal read-through transcripts that may be destabilized by either nuclear or cytoplasmic RNA quality control mechanisms (*Bresson and Tollervey, 2018*). Alternatively, a decrease in cleavage/polyadenylation activity might dampen transcription initiation, as these two processes are known to be interconnected (*Mapendano et al., 2010*). Differentiating between these possibilities will be an important next step in understanding the molecular mechanisms, which may link compromised cleavage/polyadenylation and gene expression defects in cancer. Furthermore, although we focused on annotated 3'UTR CSs in this work, similar analyses of SNVs occurring in other noncoding parts of mammalian genes (e.g. introns) might reveal an even wider impact of the loss and gain of PAS-like sequences in cancer.

In conclusion, our study reveals that the genetic inactivation of cleavage and polyadenylation in tumour suppressor genes constitutes a prevalent, yet previously overlooked category of somatic cancer mutations with driver properties. These findings emphasize the importance of pre-mRNA processing in the biology of cancer and underscore the need for improved functional annotation of single nucleotide variants in noncoding regions of the human genome.

# Materials and methods

**Key resources table**

| Reagent type (species) or resource | Designation | Source or reference | Identifiers | Additional information |
|---|---|---|---|---|
| Gene (*Homo sapiens*) | *XPA* | Ensembl | ENSG00000136936 | |
| Strain, strain background (*Escherichia coli*) | TOP10 | Thermo Fisher Scientific | Cat# C404003 | Chemically competent cells |
| Cell line (*Homo sapiens*) | Colorectal carcinoma, HCT-116 | ATCC | CCL-247 | |
| Recombinant DNA reagent | pEGFP-N3 (plasmid) | Clontech | Cat# 6080–1 | https://www.addgene.org/vector-database/2493/ |
| Recombinant DNA reagent | pGL3-Control (plasmid) | Promega | Cat# U47296 | https://www.addgene.org/212937/ |
| Recombinant DNA reagent | pML651-WT (plasmid) | This paper | Minigene | pEGFP-N3-based construct containing wild-type *XPA* 3′ region |
| Recombinant DNA reagent | pML651-MUT (plasmid) | This paper | Minigene | pEGFP-N3-based construct containing mutated *XPA* 3′ region |
| Recombinant DNA reagent | pML663-WT (plasmid) | This paper | Minigene | pGL3-based construct containing wild-type *XPA* 3′ region |
| Recombinant DNA reagent | pML663-MUT (plasmid) | This paper | Minigene | pGL3-based construct containing mutated *XPA* 3′ region |
| Sequence-based reagent | MLO4220 | This paper | PCR primers | AACGCTAGCAAATAAAGG AAATTTAGATTGGTCCT |
| Sequence-based reagent | MLO4221 | This paper | PCR primers | ATCGGTCGACTCAACAA TCAGATAGTCAACCATGA |
| Sequence-based reagent | MLO4159 | This paper | PCR primers | GCCCTAATAGCAGAGA TAAACATTGAGTTG |
| Sequence-based reagent | MLO4160 | This paper | PCR primers | CAACTCAATGTTTATCTCT GCTATTAGGGC |
| Sequence-based reagent | MLO944 | This paper | PCR primers | GGCCGCGACTCTAGATCATAA |
| Sequence-based reagent | MLO358 | This paper | PCR primers | GTAACCATTATAAGCTG CAATAAACAAG |
| Sequence-based reagent | MLO775 | This paper | PCR primers | AGAACGGCATCAA GGTGAAC |

*Continued on next page*

*Continued*

| Reagent type (species) or resource | Designation | Source or reference | Identifiers | Additional information |
|---|---|---|---|---|
| Sequence-based reagent | MLO776 | This paper | PCR primers | TGCTCAGGTAGTG GTTGTCG |
| Commercial assay or kit | jetPRIME transfection reagent | Polyplus | Cat# 101000015 | |
| Commercial assay or kit | Dual-Glo Luciferase Assay System | Promega | Cat# E2920 | |
| Software, algorithm | APARENT2 | *Linder et al., 2022* | | https://github.com/johli/aparent-resnet |
| Software, algorithm | DIGDriver | *Sherman et al., 2022* | | https://github.com/maxwellsh/DIGDriver |
| Software, algorithm | cluster Profiler | *Wu et al., 2021* | RRID:SCR_016884 | http://bioconductor.org/packages/release/bioc/html/clusterProfiler.html |

## Source data sets

Pre-mRNA CS positions and the corresponding metadata were obtained from the PolyA_DB3 database (*Wang et al., 2018*) (release 3.2 https://exon.apps.wistar.org/PolyA_DB/). The phase-3 1000 genomes vcf files were downloaded from the International Genome Sample Resource (https://ftp.1000genomes.ebi.ac.uk/vol1/ftp/data_collections/1000_genomes_project/release/). Cancer somatic SNVs and indels from whole-genome sequencing of 2583 unique tumours (PCAWG) were downloaded from the International Cancer Genome Consortium (ICGC) data portal (https://dcc.icgc.org/) and the database of Genotypes and Phenotypes (dbGaP) (project code: phs000178). Only bona fide SNVs that differed from the reference genome at a single-nucleotide position were included in the analysis. The v97 release of the Cancer Gene Census was downloaded from https://cancer.sanger.ac.uk/cosmic/download.

## Data processing

CSs located in 3'UTRs according to PolyA_DB3 were extended by 102 nt on both sides to generate 205-nt intervals. All SNVs from the 1000 genomes and the PCAWG datasets mapping to these intervals were kept for further analyses (paSNVs). FASTA files corresponding to wild-type and mutant 205-nt intervals were analysed by the APARENT2 (*Linder et al., 2022*). For each variant, we estimated the log odds ratio (LOR) of mutant (mut) variant isoform abundance with respect to the wild-type (wt) abundance (abundance was calculated by summing all cleavage probabilities mapping to 205-nt interval) as follows:

$$LOR = ln\left(mut/\left(1 - mut\right)\right) - ln\left(wt/\left(1 - wt\right)\right)$$

Incidence of PAS hexamers was quantified using the vcountPattern function from the Biostrings R/Bioconductor package (doi:10.18129/B9.bioc.Biostrings). Evolutionary conservation was calculated for either exact SNV positions or 15-nt SNV-centred windows using the GenomicScores (*Puigdevall and Castelo, 2018*) and the phastCons100way.UCSC.hg19 (*Siepel et al., 2005*) R/Bioconductor packages. Only unique SNV entries were kept for further analysis. In cases where a single SNV was located near more than one distinct CS, the strongest effect on cleavage/polyadenylation was used for further analyses. GO terms enrichment was analysed using the ClusterProfiler R/Bioconductor package (*Wu et al., 2021*).

To analyse changes in polyadenylation scores of all mutations affecting AWTAAA sequences in 3'UTRs in *Figure 2B*, APARENT2 scores were calculated for all SNV-centred 205-nt intervals from both datasets located within canonical UCSC 3'UTRs. SNVs disrupting AWTAAA sequence with LOR≤–1 within 100-nt intervals centred around polyA_DB3 CSs were considered 'annotated'.

## Cancer Census gene enrichment

Enrichment of different types of SNVs in Cancer Census genes was calculated using a two-tailed Fisher's exact test. Somatic SNVs in protein-coding sequences were classified as 'Nonsense,' 'Missense,' or 'Synonymous' based on the information provided in PCAWG maf files ('Variant_Classification'

column). Tumour suppressors were defined as genes labelled as 'TSG' but not 'Oncogene' in the Census dataset. A similar stringent approach was used to define oncogenes. Genes annotated as both 'Tumour suppressors' and 'Oncogenes' were excluded (most analyses), analysed as 'Both' (*Figure 3A* and *Figure 3—figure supplement 1*), or combined with tumour suppressors to form the extended 'Tumour suppressor+' group (*Figure 4—figure supplement 2B*).

## DigDriver enrichment analysis

We used the 'Analyzing new mutation sets' mode of DigDriver to process different functional categories of somatic SNVs. Functional annotation was taken from DigPreprocess.py annotMutationFile output files. Enrichment/excess of mutations of the Census cancer gene category was calculated as:

$$\frac{\frac{\sum observed\ mut\ cat\,1}{\sum expected\ mut\ cat\,1}}{\frac{\sum observed\ mut\ noncancer\ genes}{\sum expected\ mut\ noncancer\ genes}}$$

To calculate the 95% confidence interval (CI) of this enrichment, we performed bootstrap resampling of tumour suppressors, oncogenes and non-cancer genes in each mutation class for 1000 iterations. In each iteration, the enrichment/excess of mutations was calculated as described above. The $2.5^{th}$ and $97.5^{th}$ percentiles of the resampled distribution were used as the 95% confidence interval boundaries.

## Gene expression analysis

To analyse the possible effect of DOWN-paSNVs on transcript abundance, we selected tumour suppressor genes from the published colorectal adenocarcinoma study *Calabrese et al., 2020*, which contained DOWN-paSNVs and no other damaging SNVs (i.e. Non-syn. variants from the binarized gene-centric table in *Calabrese et al., 2020*) in some samples, and no mutations in other samples. We normalized the available gene expression data (FPKM) to account for gene copy number variation and $\log_2$-tranformed them to obtain Log2(nFPKM) values. Gene-specific Log2(nFPKM) values for the wild-type samples were then subtracted from corresponding Log2(nFPKM) values for the DOWN-paSNVs samples to obtain distributions of gene expression differences ($\Delta$Log2(nFPKM)). A one-tailed Wilcoxon signed-rank test was used to analyse the significance of a negative shift of $\Delta$Log2(nFPKM) distributions compared to 0.

## DNA constructs

All plasmids were propagated in the TOP10 *E. coli* strain (Thermo Fisher Scientific, cat# C404003). To generate read-through *XPA* minigenes (pML651-WT and pML651-MUT), a 431-nt gBlock fragment (Integrated DNA Technologies) encoding the human *XPA* 3'UTR in its natural context (chr9:100436867–100437297; GRCh37/hg19) and either the wild-type or mutated PAS were cloned into the pEGFP-N3 plasmid (Clontech) at the BsrGI and NotI sites. To generate luciferase reporter plasmids (pML663-WT and pML663-MUT), the entire *XPA* 3'UTR (chr9: 100437071–100437680; GRCh37/hg19) was amplified from HCT-116 genomic DNA using KAPA HiFi DNA polymerase HotStart ReadyMix (Roche, cat# KK2601) with MLO4220 (5'-AACGCTAGCAAATAAAGGAAATTTAGATTGGTCCT-3') and MLO4221 (5'-ATCGGTCGACTCAACAATCAGATAGTCAACCATGA-3') primers. The PCR product was gel-purified and cloned into the pGL3-Control plasmid (Promega, cat# U47296) at the XbaI and SalI sites. The cancer-specific PAS mutation was introduced using a modified Quikchange site-directed mutagenesis protocol, using the KAPA HiFi DNA polymerase HotStart ReadyMix (Roche, cat# KK2601) with MLO4159 (5'-GCCCTAATAGCAGAGATAAACATTGAGTTG-3') and MLO4160 (5'-CAACTCAATGTT TATCTCTGCTATTAGGGC-3') primers. All constructs were verified by Sanger sequencing. Plasmid maps are available on request.

## Minigenes experiments

HCT-116 cells were purchased from ATCC (Cat# CCL-247), confirmed to have characteristic morphology using light microscopy (https://www.atcc.org/products/ccl-247#detailed-product-images), and tested negative for mycoplasma contamination using a LookOut Mycoplasma PCR Detection Kit (Sigma-Aldrich, cat# MP0035). Cells were cultured in a humidified incubator at 37 °C, 5% CO2, in DMEM

containing 4.5 g/L glucose, GlutaMAX and 110 mg/L sodium pyruvate (Thermo Fisher Scientific, cat# 11360070) supplemented with 10% FBS (Hyclone, cat# SV30160.03) and 100 units/ml PenStrep (Thermo Fisher Scientific, cat# 15140122). For passaging, cells were washed with 1x PBS and dissociated in 0.05% Trypsin-EDTA (Thermo Fisher Scientific, cat# 15400054) for 10 min at 37 °C.

For read-through minigene transfection experiments, cells were typically seeded overnight in 1 mL of culture medium at $1–2 \times 10^5$ per well of a 12-well plate. Next morning, 1 µg of plasmid DNA was mixed with 2.5 µl of jetPRIME transfection reagent (Polyplus, cat# 101000015) in 150 µl of jetPRIME transfection buffer, incubated for 10 min at RT and added drop-wise to the cells. Total RNAs were extracted from cells 24 hr post-transfection using TRIzol (Thermo Fisher Scientific, cat#15596026) with an additional acidic phenol-chloroform (1:1) extraction step. The aqueous phase was precipitated with an equal volume of isopropanol, washed with 70% ethanol, and dissolved in 80 µl of nuclease-free water (Thermo Fisher Scientific, cat# AM9939). RNA samples were then treated with 4–6 units of Turbo DNase (Thermo Fisher Scientific, cat# AM2238) at 37 °C for 30 min to remove the bulk of plasmid DNA contamination, extracted with an equal volume of acidic phenol-chloroform (1:1), precipitated with 3 vol of 100% ethanol and 0.1 vol of 3 M sodium acetate (pH 5.2), washed with 70% ethanol and rehydrated in nuclease-free water. To remove any remaining traces of DNA, the RNA samples were additionally pre-treated with two units of RQ1-DNAse (Promega, cat# M6101) per 1 µg of RNA at 37 °C for 30 min. RQ1-DNAse was inactivated by adding the Stop Solution as recommended and the RNAs were immediately reverse-transcribed using SuperScript IV (Thermo Fisher Scientific, cat# 18090050) and random decamer (N10) primers at 50 °C for 30 min. cDNA samples were analysed by qPCR using a Light Cycler96 Real-Time PCR System (Roche) and qPCR BIO SyGreen Master Mix (PCR Biosystems, cat# PB20.11–51). The RT-qPCR signals downstream of the *XPA* cleavage site (MLO944, 5'-GGCCGCGACTCTAGATCATAA-3' and MLO358, 5'-GTAACCATTATAAGCTGCAATAAACAAG-3') were normalized to those obtained using upstream primers (MLO775, 5'-AGAACGGCATCAAGGTGAAC-3' and MLO776, 5'-TGCTCAGGTAGTGGTTGTCG-3').

For luciferase minigene transfection experiments, cells were typically seeded overnight in 100 µl of culture medium at $5 \times 10^3$ per well of a 96-well plate. Next morning, 70 ng of a firefly luciferase reporter construct containing *XPA* sequences and 30 ng of the *Renilla* luciferase control (pRL-TK; Promega) were mixed with 0.2 µl of jetPRIME transfection reagent in 10 µl of jetPRIME transfection buffer, incubated for 10 min at RT and added drop-wise to the cells. Following a 24 hr incubation, transfected cells were analysed using a Dual-Glo Luciferase Assay System (Promega, cat# E2920) as recommended by the manufacturer. Luminescence was measured using a Berthold Mithras LB940 plate reader.

## Statistics

Unless stated otherwise, all statistical procedures were performed in R. Data were averaged from at least three biological replicates and shown as box plots, with box bounds representing the first and the third quartiles and whiskers extending from the first and the third quartile to the lowest and highest data points or, if there are outliers, 1.5x of the interquartile range. Data obtained from RT-qPCR and luciferase assays were compared using the two-tailed Student's t-test assuming unequal variances. Genome-wide data were analysed using the Wilcoxon rank sum test or Fisher's exact test (two-tailed if not stated otherwise). Specific tests used and the p-values obtained are indicated in the figures and/or figure legends.

## Acknowledgements

We thank Maria Andrianova, Yegor Bazykin, Tim Hubbard, and Snezhka Oliferenko for helpful discussions. The results presented here are based in part on data generated by The Cancer Genome Atlas managed by the NCI and NHGRI (https://www.cancer.gov/ccg/research/genome-sequencing/tcga). This work was supported by the Biotechnology and Biological Sciences Research Council (grants BB/V006258/1 and BB/R001049/1 to EVM).

# Additional information

## Funding

| Funder | Grant reference number | Author |
|---|---|---|
| Biotechnology and Biological Sciences Research Council | BB/V006258/1 | Eugene V Makeyev |
| Biotechnology and Biological Sciences Research Council | BB/R001049/1 | Eugene V Makeyev |

The funders had no role in study design, data collection and interpretation, or the decision to submit the work for publication.

## Author contributions

Yaroslav Kainov, Conceptualization, Data curation, Software, Formal analysis, Validation, Investigation, Visualization, Methodology, Writing – original draft, Project administration, Writing - review and editing; Fursham Hamid, Data curation, Software, Formal analysis, Visualization, Methodology, Writing – original draft; Eugene V Makeyev, Conceptualization, Supervision, Funding acquisition, Validation, Investigation, Visualization, Methodology, Writing – original draft, Project administration, Writing - review and editing

## Author ORCIDs

Yaroslav Kainov ⓘ https://orcid.org/0000-0003-0664-9102
Fursham Hamid ⓘ https://orcid.org/0000-0002-6846-7382
Eugene V Makeyev ⓘ https://orcid.org/0000-0001-6034-6896

Reviewer #1 (Public review): https://doi.org/10.7554/eLife.99040.3.sa1
Author response https://doi.org/10.7554/eLife.99040.3.sa2

---

# Additional files

## Supplementary files

• Supplementary file 1. Cancer somatic DOWN-paSNVs. This table lists somatic DOWN-paSNVs predicted to reduce cleavage and polyadenylation efficiency across various Pan-Cancer Analysis of Whole Genomes (PCAWG) samples. For more information, please refer to the Description tab.

• MDAR checklist

## Data availability

All data generated or analysed during this study are included in the manuscript and supporting files.

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
