## [Editor Report · eLife Assessment]

This **important** study substantially advances our understanding of noncoding somatic mutations by identifying a novel class of mutations that affect 3'UTR polyadenylation signals enriched in tumor suppressor genes in cancer. The evidence supporting the conclusions is **convincing**, with rigorous statistical analyses. The work will be of broad interest to cancer researchers.

---

## [Referee Report · Reviewer #1 (Public review)]

Kainov et al investigated the prevalence of mutations in 3'UTR that affect gene expression in cancer to identify noncoding cancer drivers.

The authors used data from normal controls (1000 genome data) and compared it to cancer data (PCAWG). They found that in cancer 3'UTR mutations had a stronger effect on cleavage than the normal population. These mutations are negatively selected in the normal population and positively selected in cancers. The authors used PCAWG data set to identify such mutations and found that the mutations that lead to a reduction of gene expression are enriched in tumor suppressor genes and those that are increased in gene expression are enriched for oncogenes. 3'UTR mutations that reduce gene expression or occur in TSGs co-occur with non-synonymous mutations. The authors then validate the effect of 3'UTR mutations experimentally using a luciferase reporter assay. These data identify a novel class of noncoding driver genes with mutations in 3'UTR that impact polyadenylation and thus gene expression.

This is an elegant study with fundamental insight into identifying cancer driver genes. The conclusions of this paper are mostly well supported by data, but some aspects of data analysis need to be extended.

Comments on revisions:

The authors addressed most of my comments.

---

## [Author Response]

The following is the authors’ response to the original reviews.

**Public Reviews:**

**Reviewer #1 (Public Review):**
Kainov et al investigated the prevalence of mutations in 3'UTR that affect gene expression in cancer to identify noncoding cancer drivers.The authors used data from normal controls (1000 genome data) and compared it to cancer data (PCAWG). They found that in cancer 3'UTR mutations had a stronger effect on cleavage than the normal population. These mutations are negatively selected in the normal population and positively selected in cancers. The authors used PCAWG data set to identify such mutations and found that the mutations that lead to a reduction of gene expression are enriched in tumor suppressor genes and those that are increased in gene expression are enriched for oncogenes. 3'UTR mutations that reduce gene expression or occur in TSGs cooccur with non-synonymous mutations. The authors then validate the effect of 3'UTR mutations experimentally using a luciferase reporter assay. These data identify a novel class of noncoding driver genes with mutations in 3'UTR that impact polyadenylation and thus gene expression.This is an elegant study with fundamental insight into identifying cancer driver genes. The conclusions of this paper are mostly well supported by data, but some aspects of data analysis need to be extended.

We thank the reviewer for the positive assessment of our work and constructive comments.

(1) It would be important for the authors to show if the findings of this study hold for metastatic cancers since most deaths occur due to metastasis and tumor heterogeneity changes when cancer progresses to metastasis. The authors should use the Hartwig data and show if metastatic cancers are enriched for 3'UTR mutations.

This is a good suggestion, but we believe that the proposed analysis would have a significantly stronger impact in the context of a separate study focused specifically on longitudinal changes in the somatic mutation landscape as cancer progresses from primary tumours to metastases. Conducting such a study would require obtaining permissions to use relevant controlled datasets and, ideally, collaborating with oncologists to generate additional genome and transcriptome sequencing data. As such, this level of analysis would go beyond the current scope of our work.

(2) Figure 2 should show the distribution of 3'UTR mutations by cancer type especially since authors go on to use colorectal cancer only for validations. It would be helpful to bring Figures S3A and S3C to this panel since these findings make the connections to cancer biology. Are any molecular functions enriched in addition to biological processes? Are kinases, phosphatases, etc more or less affected by 3'UTR mutations?

As suggested, we have added a pie chart showing the distribution of 3’UTR mutations by cancer type (new Fig. 2E). Notably, nearly a half of the mutations in our dataset was of colorectal adenocarcinoma origin, justifying the focus on this type of cancer in our subsequent validation analyses.

To strengthen the connections to cancer biology, we moved Fig. S3A and S3C to the main text. It was more logical to integrate these panels into Fig. 3 rather than Fig. 2. We also analysed molecular function enrichment in Fig. 3E. Consistent with the biological process enrichment (now shown in Fig. 3D), this revealed an enrichment of proteins interacting with the ubiquitination pathway, including tumour suppressors *SMAD2*, *APC* and *AXIN1*.

(3) Figure 3 looks at the co-occurrence of 3'UTR mutations with non-synonymous mutations but what about copy number change? You would expect the loss of the other allele to be enriched. Along the same line, are these data phased? Do you know that the nonsynonymous mutations are in the other allele or in the same allele that shows 3'UTR mutation?

As suggested, we have analysed copy number variation data. As mentioned in the revised Results, this "showed that increased copy number was 4.1-times more common in the PCAWG data compared to allele loss. However, the incidence of copy number increase was substantially lower in the DOWN-paSNV group compared to the BG-paSNV control (Fig. S6). This points to a negative selection against duplications of genes affected by DOWNpaSNVs in cancer".

Phasing somatic mutations in cancer samples is challenging due to high genetic heterogeneity of tumour cells. This situation will likely improve in the near future with the increased use of long-read sequencing. However, with currently available data, there is no straightforward method to determine whether mutations co-occur in the same cell. We have added a note on this in the Discussion section: "As long-read genomic sequencing data become increasingly available, it will be interesting to investigate whether these additional mutations occur in the same or in a different allele compared to the DOWN-paSNVs".

**Reviewer #2 (Public Review):**
Summary:To evaluate whether somatic mutations in cancer genomes are enriched with mutations in polyadenylation signal regions, the authors analyzed 1000 genomes data and PCAWG data as a control and experimental set, respectively. They observed increased enrichment of somatic mutations that may affect the function of polyA signals and confirmed that these mutations may influence the expression of the gene through a minigene expression experiment.Strengths:This study provides a systematic evaluation of polyA signal, which makes it valuable. Overall, the analytic approach and results are solid and supported by experimental validation.

Thank you.

Weaknesses:(1) This study uses APARENT2 as a tool to evaluate functional alteration in polyA signal sequences. Based on the original paper and the results shown in this paper, the algorithm appears to be of high quality. However, the whole study is dependent on the output of APARENT2. Therefore, it would be nice to(a) run and show a positive control run, which can show that the algorithm works well, and (b) describe the rationale for selecting this algorithm in the main text.

As suggested, we have added control analyses to Fig. S1A-B, which show that APARENT2 performs well in our hands. We have described the rationale for using APARENT in the Results as follows: "For each paSNV, we calculated the change in cleavage/polyadenylation efficiency using the APARENT2 neural network model, which has been shown to infer this statistic more accurately than earlier approaches [Ref23]".

(2) Are there recurrent somatic mutation calls (=exactly the same mutation across different tumor samples) in the poly(A) region of certain genes?

We indeed see several cases where the same cleavage/polyadenylation signal is affected by the same or different DOWN mutations in different cancer samples. This finding is now summarized in the Results section and Table S1 as follows: "In several cases, including *LRP1B* and *FOXO1*, which are known to act as tumour suppressors in certain cancers, the same signal/polyadenyalation signal was disrupted by the same or different mutations in more than one sample (see columns Mut_Recurrence and Signal_Recurrence in Table S1)".

(3) The authors nicely showed that the minigene with A>G mutation altered gene expression. Maybe one can reach a similar conclusion by analyzing a cancer dataset that has mutation and gene expression data? That is, genes with or without polyA mutations show different expression levels.

The data presented in Fig. 5A-B show that DOWN-paSNV mutations have a negative effect on the expression of endogenous tumour suppressor genes.

**Recommendations for the authors:**

**Reviewer #1 (Recommendations For The Authors):**
Figures should be numbered in order. For example, Figure S3C is referred to in the text before S3A-B, etc.

We have proofread the text to fix this problem.

Adding a supplementary file with lists of genes carrying 3'UTR mutations split by effect on gene expression and cancer type would be very useful for the community.

We now show this in Table S1, with the caveat that we could not consistently investigate the effect of DOWN-paSNV on gene expression since the transcriptomics data are not available for all cancers.

Spelling mistake in Figure 1A - genone should be genome.

Fixed - thank you.

Typo in Figure 1B x-axis label +50nt should be -50nt to the left of the dashed line.

Fixed - thank you.

All figures use E to denote x10 but it would make the figures more readable if authors used the standard notation (x10) for all numbers with exponents and base 10.

Done.